# Methionine Promotes Milk Protein Synthesis via the PI3K-mTOR Signaling Pathway in Human Mammary Epithelial Cells

**DOI:** 10.3390/metabo13111149

**Published:** 2023-11-14

**Authors:** Peizhi Li, Xibi Fang, Guijie Hao, Xiaohui Li, Yue Cai, Yuhao Yan, Liting Zan, Runjun Yang, Boqun Liu

**Affiliations:** 1Laboratory of Nutrition and Functional Food, College of Food Science and Engineering, Jilin University, Changchun 130062, China; pzli21@mails.jlu.edu.cn (P.L.); yhyan22@mails.jlu.edu.cn (Y.Y.); zanliting2022@163.com (L.Z.); 2College of Animal Science, Jilin University, Changchun 130062, China; xibifang@jlu.edu.cn; 3Key Laboratory of Healthy Freshwater Aquaculture, Ministry of Agriculture and Rural Affairs, Huzhou 313001, China; haoguijie202211@163.com; 4Huzhou Key Laboratory of Aquatic Product Quality Improvement and Processing Technology, Zhejiang Institute of Freshwater Fisheries, Huzhou 313001, China; 5Center of Animal Experiment, College of Basic Medical Sciences, Jilin University, Changchun 130021, China; lixiaohui@jlu.edu.cn; 6HaMi Inspection and Testing Center, Hami 839000, China; rebeccaccy@163.com

**Keywords:** MCF-10A cells, methionine, milk synthesis, mTOR

## Abstract

Breast milk is widely considered to be the most natural, safe, and complete food for infants. However, current breastfeeding rates fall short of the recommendations established by the World Health Organization. Despite this, there are few studies that have focused on the promotion of human lactation through nutrient supplementation. Therefore, the aim of this study was to investigate the effect of methionine on milk synthesis in human mammary epithelial cells (MCF-10A cells) and to explore the underlying mechanisms. To achieve this, MCF-10A cells were cultured with varying concentrations of methionine, ranging from 0 to 1.2 mM. Our results indicated that 0.6 mM of methionine significantly promoted the synthesis of milk protein. An RNA-seq analysis revealed that methionine acted through the PI3K pathway. This finding was validated through real-time quantitative polymerase chain reaction (RT-qPCR) and Western blotting. In addition, PI3K inhibition assays confirmed that methionine upregulated the expression of both mTOR and p-mTOR through activation of PI3K. Taken together, these findings suggest that methionine positively regulates milk protein synthesis in MCF-10A cells through the PI3K-mTOR signaling pathway.

## 1. Introduction

Breastfeeding and breast milk constitute the pinnacle of standards for fostering infants’ optimal growth and developmental trajectory [1]. In 2012, the World Health Assembly (WHA) issued the “Global Nutrition Goals 2025”, encompassing a range of aspirations. One of the goals is to improve breastfeeding and increase exclusive breastfeeding rates to 50%, by 2025, for infants aged 0–6 months. This goal is underpinned by multifaceted rationales. Primarily, breast milk substantiates its nutritional supremacy by endowing neonates with a repository of bioactive constituents, prominently including amino acids, peptides, and proteins [2]. Compared to commercial infant formula, human milk boasts a slightly augmented content of free amino acids [3]. Beyond its nutritional role, there are various proteins with immunomodulatory activities in breast milk, such as immunoglobulin A (IgA), immunoglobulin M (IgM), immunoglobulin G (IgG), and lactoferrin. Additionally, breast milk is also rich in lipids, carbohydrates, growth-promoting factors, immune factors, and vitamins [4]. Compared to cow’s milk, breast milk is rich in oligosaccharides [5]. The distinct attributes of oligosaccharides in breast milk are intrinsically linked to their constructive impact on early immune function in neonates, coupled with their role in shaping the maturation trajectory of the gut microbiota [6].

Breast milk shows a multitude of multifunctional attributes. Notably, breast milk manifests a pronounced anti-inflammatory effect upon neonates, attributed to the orchestrated actions of transformative growth factor-β, interleukin-10, erythropoietin, and lactoferrin. These constituents collectively contribute to a robust anti-inflammatory milieu, functioning synergistically or individually [7]. Evidential observations underscore breastfeeding’s role as a protective factor against an array of maladies encompassing infectious, atopic, and cardiovascular disorders. The mounting body of evidence further suggests that breastfeeding potentially exerts a safeguarding influence against future occurrences of overweight and diabetes [8]. Elucidating the intricate interplay between breastfeeding and cognitive development, research has substantiated a positive correlation between breastfeeding duration and children’s intelligence [9], Specifically, offspring breastfed for over 6 months have been shown to exhibit a slightly elevated cognitive quotient compared to those breastfed for less than 6 months [10]. Moreover, a prolonged breastfeeding duration translates into reduced incidence rates of infectious ailments and associated mortality among children [11]. This is of great significance to the healthy growth of a baby. Therefore, breastfeeding can promote rapid growth of babies under better nutritional conditions and safety protection. In short, breast milk is expected to have different functions in addition to nutrition.

Nonetheless, extensive investigations have underscored a concerning reality, i.e., the prevlence of exclusive breastfeeding falls below anticipated levels in numerous nations [12]. It is reported that, in 2020, only 44% of infants were exclusively breastfed up to 6 months of age. The duration of breastfeeding in high-income countries is lower than in resource-poor countries [12]. Furthermore, breastfeeding rates in middle-income and common-income countries persistently lag behind the recommendations posited by the World Health Organization (WHO). Particularly noteworthy is the subpar performance observed in the South-East Asia/West Pacific regions and among middle-income and common-income countries, notably in relation to the introduction of complementary feeding between 6 and 8 months of age [13]. In China, the rate of exclusive breastfeeding in the first six months is still low, and the most significant factor for the low feeding rate is the lack of breast milk [14].

Currently, the research landscape concerning nutrient-driven enhancement of human lactation remains notably deficient. The majority of prior investigations into nutrient supplementation for lactation have predominantly revolved around elevating milk production in livestock species like cows, goats, and sows [15]. Amino acids that have shown potential in augmenting milk production encompass methionine, leucine, valine, and lysine [16,17,18,19]. Among them, the potential significance of methionine as a crucial “nutritional signal” in protein metabolism has been extensively discussed [20]. Methionine is one of the limiting amino acids in the diet of dairy cows, so in previous studies, methionine has been widely used as a nutritional supplement in milk production [21,22]. Additionally, methionine supplementation has been added to goat feed to promote milk synthesis [23]. Concurrently, the utilization of 2-hydroxy-4-(methylthio)butyric acid has shown promising results in improving milk protein production and optimizing nitrogen utilization efficiency [24].

However, it is worth noting that there is currently a dearth of research exploring the potential of methionine to promote human milk protein synthesis. Therefore, we investigated the effects of methionine on human mammary epithelial cell (MCF-10A cells were used) proliferation and milk protein synthesis, and then explored the molecular mechanisms by which methionine acts.

## 2. Materials and Methods

### 2.1. Cell Culture and Treatments

The MCF-10A cell line was purchased from the Cell Resource Center, IBMS, CAMS/PUMC (Beijing, China). The MCF-10A cells were cultured in Dulbecco’s Modified Eagle Medium/Nutrient Mixture F-12 (DMEM/F-12) (Corning, Corning, NY, USA) supplemented with 10 μg/mL of insulin (Solarbio, Beijing, China), 0.5 μg/mL of hydrocortisone (Solarbio, Beijing, China), 20 ng/mL of epidermal growth factor (EGF) (Peprotech, Rocky Hill, NJ, USA), 100 ng/mL of cholera toxin (Absin, Shanghai, China), and 10% fetal bovine serum (FBS) (Tianhang Biotechnology, Huzhou, China) under the condition of 37 °C and 5% CO_2_. The MCF-10A cells reached 80–90% convergence and were treated with 0.05% trypsin (Thermo Fisher Scientific, Waltham, MA, USA). To detect the effect of methionine on the MCF-10A cells, they was cultured in six-well plates (Biofil, Guangzhou, China). When the cells converged to 70%, the medium was changed to a serum-free medium. The MCF-10A cells were induced by the addition of prolactin (Y-S Biotechnology, Shanghai, China) and treated with different concentrations of methionine (0, 0.3, 0.6, 0.9, and 1.2 mM) for 24 h. To observe the effect of PI3K inhibition, XL-147 (KKL MED, Ashland, OR, USA) (477 nM) was added 2 h before the indicated treatments.

### 2.2. Cell Viability

To detect the effect of methionine concentration on cell viability, the MCF-10A cells were cultured in 96-well culture plates (Biofil, Guangzhou, China). After the cell treatment was completed, 10 μL of CCK8 reagent (Beyotime, Shanghai, China) was added to each well and incubated for 0.5 h. The absorbance values were obtained at 450 nm using an enzyme marker (YongChuang SM600, Shanghai, China), and cell viability was calculated as follows:$$\mathrm{Cell}\;\mathrm{viability}=\frac{\mathrm{Absorbance}\;\mathrm{value}\;\mathrm{of}\;\mathrm{the}\;\mathrm{experimental}\;\mathrm{group}\;-\;\mathrm{Absorbance}\;\mathrm{value}\;\mathrm{of}\;\mathrm{the}\;\mathrm{blank}\;\mathrm{group}}{\mathrm{Absorbance}\;\mathrm{value}\;\mathrm{of}\;\mathrm{the}\;\mathrm{control}\;\mathrm{group}\;-\;\mathrm{Absorbance}\;\mathrm{value}\;\mathrm{of}\;\mathrm{the}\;\mathrm{blank}\;\mathrm{group}}$$

For the experimental group, the MCF-10A cells were treated with 0.3, 0.6, 0.9, and 1.2 mM of methionine respectively.

For the control group, the MCF-10A cells were treated without methionine.

### 2.3. Measurement of Cell Proliferation

To detect the effect of methionine concentration on cell proliferation, the MCF-10A cells were cultured in 24-well culture plates (Biofil, Guangzhou, China). The proliferated cells were detected using an EdU Kit (Beyotime, Shanghai, China). The staining of the cells was observed with a fluorescent microscope (Nikon TE2000, Tokyo, Japan). Three microscopic fields of view were randomly selected for each well. The proliferated cells were shown with green fluorescence, and nuclei were shown with blue fluorescence. We used the ImageJ software (ImageJ 2, National Institutes of Health, Bethesda, MD, USA) for the cell count analysis.

### 2.4. Immunofluorescence Assay

The MCF-10A cells with appropriate density were disseminated to a glass-bottom cell culture dish (Nest, Wuxi, China). After treatment with methionine, cell density reached 60%. The cells were fixed in 4% paraformaldehyde (Biosharp, Hefei, China) for 22 min, and then permeabilized with 0.1% Triton-X100 (Bio Basic, Toronto, ON, Canada) for 5–10 min, followed by closure with 5% bovine serum albumin (BSA) (Sigma-Aldrich, Darmstadt, Germany) for 30 min at 4 °C. The cells were covered with primary antibody beta casein (CSN2) (Proteintech, Wuhan, China) overnight at 4 °C. After washing three times with PBST, the cells were incubated with fluorescein isothiocyanate (FITC)-labeled secondary antibody (Proteintech, Wuhan, China) at 37 °C, in the dark, for 2 h. Finally, the cells were incubated with Hoechst (Beyotime, Shanghai, China) in the dark for 10 min. Protein expression was observed by fluorescence microscopy (Nikon TE2000, Tokyo, Japan).

### 2.5. RNA Extraction and Real-Time Quantitative PCR

Total RNA from the MCF-10A cells was extracted using a FastPure Cell/Tissue Total RNA Isolation Kit (Vazyme, Nanjing, China), and the concentration of RNA was measured using a spectrophotometer (NanoDrop 2000, Thermo Fisher Scientific, Waltham, MA, USA). The cDNA was synthesized using a reverse transcription kit (Tiangen Biotech, Beijing, China) for 1 μg of total RNA. To detect the relative expression levels of mRNA of genes by real-time quantitative polymerase chain reaction (RT-qPCR), primers for RT-qPCR were designed using the Primer Premier 6.0 software (Premier Biosoft International, CA, USA). RT-qPCR primers of CSN2, casein kappa (CSN3), casein alpha S1 (CSN1S1), fatty acid desaturase 3 (FADS3), solute carrier family 16 member 4 (SLC16A4), mitogen-activated protein kinase 10 (MAP3K10), interleukin-33 (IL33), integrin subunit beta 2 (ITGB2), sterile alpha motif domain containing 9 like (SAMD9L), and β-actin (an internal reference gene) are shown in Table 1. RT-qPCR reaction system: forward primer 0.2 μL (10 μmol/L), reverse primer 0.2 μL (10 μmol/L), SYBR Green Real-Time PCR Master Mix (ABclonal, Wuhan, China) 5 μL, cDNA 1 μL, and ddH2O 3.6 μL. Reaction conditions: 95 °C for 3 min; 95 °C for 5 s, 60 °C for 30 s, 40 cycles. The relative mRNA expression of target genes was measured using the 2^−ΔΔCt^ method.

### 2.6. Western Blotting

After convergence of the MCF-10A cells to 90%, they were lysed in cold RIPA lysis buffer (Beyotime, Beijing, China). Then, the lysate was cooled on ice for 5 min and shaken for another 5 min. The procedure was repeated 5 times. Finally, the lysate was centrifuged at 4 °C with 12,000× *g* for 15 min. The protein samples were electrophoresed in sodium dodecyl sulfate (SDS) polyacrylamide gels. Next, the proteins were transferred to polyvinylidene fluoride and incubated with the indicated primary antibody. The primary antibodies were β-actin (Cell Signaling Technology, Wikiwand, MA, USA); CSN2 (Proteintech, Wuhan, China); mTOR (mammalian target of rapamycin) (Proteintech, Wuhan, China); p-mTOR (Proteintech, Wuhan, China); PI3K (phosphoinositide 3-kinase) (Proteintech, Wuhan, China). Finally, the samples were incubated with the corresponding secondary antibodies: Goat Anti-Mouse IgG H&L/HRP (Bioss, Beijing, China) and Goat Anti-Rabbit IgG (H+L) HRP (Bioworld, Saint Paul, MN, USA). Western blotting reagents were visualized with chemiluminescence (ECL) (Beyotime, Beijing, China) and quantified using the ImageJ software (ImageJ 2, National Institutes of Health, Bethesda, MD, USA).

### 2.7. RNA-Seq Analysis

Library preparation and gene sequencing were commissioned by Shanghai Applied Protein Technology Co. (Shanghai, China). Total RNA was extracted using the TRIzol method. After the sample was tested, eukaryotic mRNA was enriched with magnetic beads with Oligo (dT). Then, the mRNA was randomly interrupted by the addition of fragmentation buffer. The first strand of cDNA was synthesized using six-base random hexamers with mRNA as the template. The second strand of cDNA was synthesized by adding buffer, dNTPs, and DNA polymerase I. Then, the double cDNA was purified using AMPure XP beads. Then, the purified double-stranded cDNA was end-repaired, A-tailed, and sequenced, followed by fragment size selection using AMPure XP beads, and finally PCR enrichment to obtain the final cDNA library. After passing the assay, the libraries were pooled and sequenced according to the target sequencing data volume.

### 2.8. Statistical Analysis

The results are expressed as means ± standard error of measurement (SEM). Significance was analyzed by one-way analysis of variance (ANOVA) and Tukey’s multiple comparisons. *p* < 0.05 was defined as statistically significant.

## 3. Results

### 3.1. Effects of Methionine on the Proliferation of MCF-10A Cells

The effect of methionine on the proliferation of MCF-10A cells was determined. The CCK-8 assay results are shown in Figure 1A. The results revealed that the addition of methionine within the range of 0–1.2 mM did not exhibit any cytotoxic effects on the MCF-10A cells. Furthermore, the cell viability was observed to be higher at 24 h compared to 48 h following methionine treatment. To further investigate the proliferative effects of methionine, the EdU experiments were conducted (Figure 1B,C). The results demonstrated that cell proliferation initially increased, and then decreased after 24 h of treatment with varying concentrations of methionine. Notably, the proliferation of cells was significantly enhanced with the 0.3 mM and 0.6 mM methionine treatments (*p* < 0.05).

### 3.2. Effects of Methionine on Milk Protein Synthesis in MCF-10A Cells

We proceeded to investigate the influence of methionine on genes implicated in casein synthesis. We assessed the mRNA expression levels of CSN2, CSN3, and CSN1S1 by RT-qPCR (Figure 2A–C). The findings confirmed that the methionine treatment led to an upregulation of these genes, indicating its role in promoting casein synthesis. Notably, the highest level of upregulation was observed at a concentration of 0.6 mM methionine. To further validate these findings, the protein expression levels of CSN2 were assessed using Western blotting (Figure 2D,E). The results demonstrated a dose-dependent increase in CSN2 expression with the addition of methionine, with the peak increase observed at 0.6 mM of methionine. These Western blotting results were consistent with the mRNA expression levels of CSN2. Immunofluorescence showed that cell morphology was not altered by the addition of methionine (Figure 2F), indicating that methionine had no toxic effect on the cells. It was also evident that CSN2 was detectable in both the nucleus and the cytoplasm and was evenly distributed in the cytoplasm, strong in the nucleus, and weak in the cytoplasm (cell membranes are uncertain). These results suggest that methionine can promote the synthesis of milk protein.

### 3.3. Analysis of DEGs in the Methionine-Supplemented and Control Groups

To elucidate the mechanism by which methionine promotes milk protein synthesis, we conducted a transcriptome analysis on cells treated with either no methionine or 0.6 mM of methionine. A total of 165 differentially expressed genes (DEGs) were identified, with 67 genes showing downregulation and 98 genes showing upregulation compared to the control group (false discovery rate (FDR) < 0.05, |log2(foldchange)| > 1) (Figure 3A). The heat map in Figure 3B illustrates the clustering of samples based on similar gene expression patterns in response to treatment. The top 15 DEGs are listed in Table 2. Among these genes, IL33, ITGB2, ITGB3, and ITGA10 are associated with immunity [25,26,27]. FADS3 is closely associated with milk fat synthesis [28].

To validate the accuracy of the RNA-seq results, we selected 6 genes (FADS3, SLC16A4, MAP3K10, IL33, ITGB2, and SAMD9L) among the 165 differential genes for RT-qPCR validation. As depicted in Figure 3C, the RT-qPCR results were consistent with the RNA-seq results, demonstrating upregulation of all six genes compared to the control group. This validation confirms the accuracy and reliability of the transcriptome data obtained.

### 3.4. DEGs Participation in Biological Processes

To speculate on the pathways of gene enrichment, we used Gene Ontology (GO) and the Kyoto Encyclopedia of Genes and Genomes (KEGG) for analysis. GO annotations are used to identify biological processes (BPs), molecular functions (MFs), and cellular components (CCs). In our study, the prominent biological processes observed included the regulation of cell motility, regulation of locomotion, regulation of cellular component movement, and biological adhesion. The primary molecular functions identified were sulfur compound binding and heparin binding. Additionally, the key cellular components found to be involved were cell–cell junctions (Figure 4A). In addition, the DEGs were analyzed using the KEGG database, and the top 20 enriched pathways are shown in Figure 4B. Five of the pathways were significantly enriched (*p* < 0.05), including extracellular Matrix (ECM)–receptor interaction, the PI3K-Akt signaling pathway, glycosaminoglycan biosynthesis-keratan sulfate, phagosome, and focal adhesion. The significantly enriched DEG pathway is mainly present in environmental information processing. Therefore, the above results suggest that methionine may promote the synthesis of downstream genes through the PI3K-AKT signaling pathway, thereby affecting milk protein synthesis in MCF-10A cells.

### 3.5. Effects of Methionine on the PI3K-mTOR Signaling Pathways in MCF-10A Cells

To further explore the specific pathway by which methionine promotes milk protein synthesis, we identified a gene downstream of PI3K that is closely associated with lactation, i.e., mTOR. We performed Western blotting to assess the protein expression levels of mTOR and PI3K in the MCF-10A cells. As shown in Figure 5A–C, the methionine-treated group exhibited increased levels of mTOR and PI3K proteins compared to the control group. To confirm whether the upregulation of lactoprotein-related gene synthesis in MCF-10A cells was mediated by PI3K, we treated the cells with a PI3K inhibitor (XL-147). The results demonstrated that the addition of XL-147 led to a reduction in the mRNA expression of the three caseins (Figure 5D–F). Interestingly, simultaneous treatment of cells with both XL-147 and methionine restored the expression of these genes. Furthermore, we investigated the protein expression levels of CSN2, mTOR, and p-mTOR in response to XL-147 treatment. As depicted in Figure 5G–L, the addition of XL-147 resulted in a decrease in the protein expression levels of CSN2, mTOR, and p-mTOR. However, upon the addition of 0.6 mM of methionine, the expressions of these proteins were restored. These findings suggest that methionine activates mTOR through the PI3K pathway, thereby influencing the expression of casein-related genes involved in milk protein synthesis.

## 4. Discussion

Methionine, an essential amino acid for humans, plays a pivotal role in various physiological processes [29], including protein synthesis, DNA methylation [30], and DNA polyamine synthesis [31]. In the realm of animal husbandry, methionine often supplements feed, as research has demonstrated its efficacy in enhancing milk production and improving dairy quality in cows [32]. At the molecular level, methionine is also incorporated into bovine mammary epithelial cells (BMECs). Studies have indicated that it fosters BMEC proliferation and casein synthesis through several established pathways, such as the SNAT2-PI3K signaling pathway [18], the FABP5-SREBP-1c signaling pathway [33], and the ASCT2/SARS/mTOR signaling pathway [34]. However, the application of methionine to MCF-10A cells has not been previously explored and its underlying mechanism remains undiscovered.

Our research corroborates that the addition of methionine stimulates cell proliferation and β-casein synthesis, achieving optimal results at an addition level of 0.6 mM. Based on both our study and previous research, it has been observed that the synthesis of cellular milk proteins and cell proliferation gradually decline as the concentration of methionine increases beyond 0.6 mM. This implies that excessive methionine may possess cytotoxic properties. Previous studies have reported that at elevated levels, methionine is converted into a harmful intermediate called homocysteine [35,36]. Furthermore, excessive methionine can induce methionine transamination, which has been associated with hepatocyte toxicity in mice [37]. These findings collectively indicate that an excess of methionine may generate specific toxic substances that impede cellular growth.

Among the DEGs identified through RNA sequencing, several are notably associated with milk production. For instance, FADS3 is linked to the fatty acid composition of breast milk. Prior research has highlighted the FADS family as a crucial determinant of docosahexaenoic acid (DHA) levels in infants. Notably, the intake of DHA from breastfeeding significantly influences the erythrocyte DHA status in infants [38]. MAP3K10, a member of the serine/threonine kinase family, is implicated in cell proliferation and apoptosis [39]. Previous reports have suggested that MAP3K10 responds to amino acid stimulation. Methionine, in particular, necessitates MAP3K10 to stimulate the phosphorylation of GlyRS and NFκB1. Both of these are involved in the expression of genes associated with cellular anabolism and the proliferation of BMECs [40]. SLC16A4′s function is connected to mitochondrial respiration, basal glycolysis, and glycolytic capacity [41]. These findings suggest that the addition of methionine may influence cell proliferation and fatty acid content. However, the underlying regulatory mechanisms warrant further in vivo analysis.

In our study, we performed a KEGG pathway enrichment analysis of differentially expressed genes (DEGs) between the methionine and control groups. The analysis revealed significant enrichment in the ECM–receptor interaction and PI3K-Akt signaling pathways. The ECM–receptor interaction pathway plays a crucial role in immunomodulation, providing structural support for normal physiological activities of tissue cells. Its rich protein composition and active molecules all contribute to its immunomodulatory function [42]. This also coincides with the DEGs screened out. In previous studies, it has been shown that, compared to non-lactating mammary gland cells, lactation-derived mammary gland cells have upregulated genes associated with immune regulation response [43]. Notably, integrins, which are major ECM receptors, have been shown to facilitate several intercellular interactions [44]. Previous reports have highlighted the role of integrins in regulating the differentiation of mammary epithelial cells (MECs). Specifically, in lactating mammary epithelial cells, integrin-linked kinases have been identified as crucial connectors in integrin signal transduction. These kinases activate Rac1, which is essential for the induction of milk protein expression in response to prolactin [45]. Moreover, integrins play a regulatory role in breast acinar morphogenesis [46]. The PI3K-AKT pathway is an intracellular signaling pathway that responds to extracellular signals to promote metabolism, proliferation, cell survival, growth, and angiogenesis [47]. Previous research has demonstrated that methionine enhances milk protein and lipid synthesis and cell proliferation in bovine mammary epithelial cells via the SNAT2-PI3K signaling pathway [18]. Moreover, leucine has been shown to stimulate milk synthesis in bovine mammary epithelial cells through the PI3K-DDX59 signaling axis [19]. Based on these findings, we hypothesize that methionine may promote milk protein synthesis through involvement in the PI3K signaling pathway. Further research is needed to validate this hypothesis and unravel the precise mechanisms involved.

Our experimental findings confirmed our initial hypotheses through the application of PI3K inhibition tests. We found that methionine activates the mTOR pathway via PI3K and also impacts casein expression. The mTOR, a mammalian target protein of rapamycin, is a phosphatidylinositol kinase. It is widely distributed in cells and plays an important role not only as a signaling factor but also in cell growth and protein synthesis [48]. Previous research has indicated that amino acids can stimulate the mTOR signaling pathway, coordinating the translation process involved in milk protein synthesis [49]. When specific concentrations of lysine and methionine were introduced to bovine mammary epithelial cells, there was an activation of the mTOR signaling pathway expression [50]. The optimal amino acid ratio stimulated β-casein expression, and this was found to correlate with the mTOR pathway [51]. These findings align with the results of our study.

## 5. Conclusions

Our study offers a comprehensive analysis that underscores the significant role of methionine as a potent orchestrator of cellular proliferation and milk protein synthesis. We demonstrate that methionine acts as a positive regulator, fostering both cell growth and milk protein synthesis. When applied in appropriate concentrations, methionine significantly enhances the proliferation of MCF-10A cells and the synthesis of milk protein. The underlying mechanism of this action was further unraveled, revealing its operation via the PI3K-mTOR signaling pathway. This research represents the first thorough examination of the potential mechanisms by which methionine enhances milk protein synthesis in MCF-10A cells. It provides a robust theoretical foundation for the utilization of methionine products to promote lactation, thereby offering a novel avenue for lactation enhancement strategies.

## Figures and Tables

**Figure 1 metabolites-13-01149-f001:**
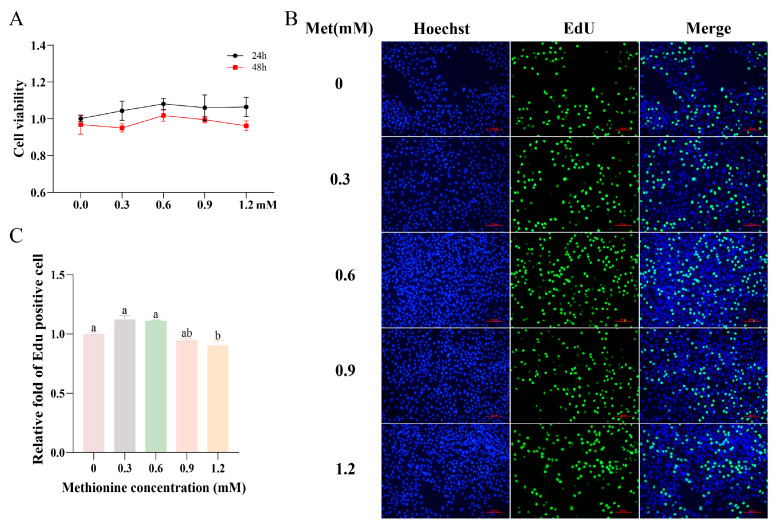
Effects of methionine on the proliferation of MCF-10A cells. MCF-10A cells were treated with different concentrations of methionine (0, 0.3, 0.6, 0.9, and 1.2 mM): (**A**) CCK8 determines the effect of different concentrations of methionine on cell viability at 24 h and 48 h (*p* > 0.05); (**B**) effects of different concentrations of methionine on the proliferation of cells using EdU staining at 24 h (nuclei were stained with Hoechst in blue, proliferative cells were labeled in green fluorescence, scale bar is 100 μm); (**C**) quantification of the EdU results. Data are reported as means ± SEM. All data were analyzed using ordinary one-way ANOVA multiple comparisons and Tukey’s multiple comparison test. Values with different superscripted lowercase letters demonstrate a significant difference (*p* < 0.05). The Met in the graphs all represents methionine.

**Figure 2 metabolites-13-01149-f002:**
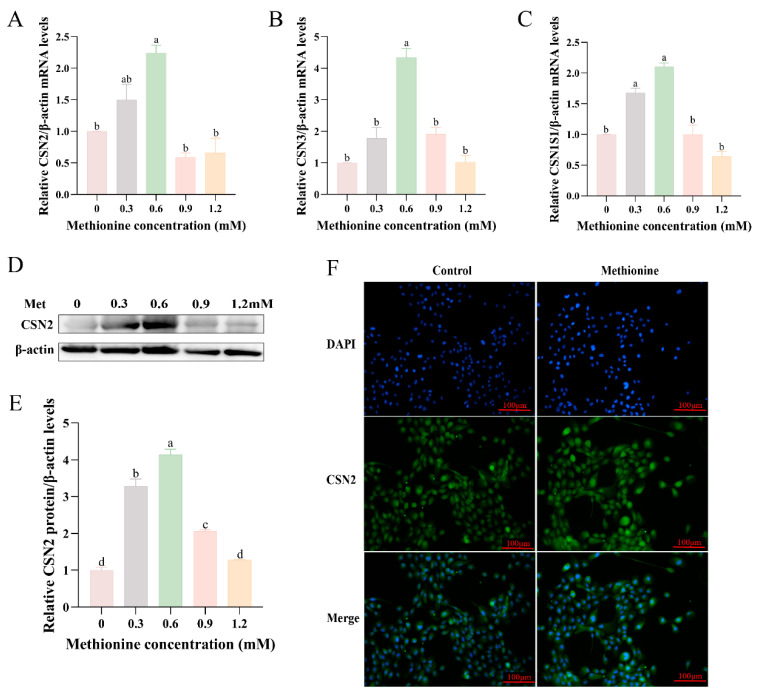
Effect of methionine on the expression of genes related to milk protein synthesis. MCF-10A cells were treated with different concentrations of methionine (0, 0.3, 0.6, 0.9, and 1.2 mM) for 24 h: (**A**–**C**) Effect of different concentrations of methionine on the mRNA expression levels of CSN2 (**A**), CSN3 (**B**), CSN1S1 (**C**); (**D**) CSN2 protein levels were measured by Western blotting analysis; (**E**) relative folds of β-casein (CSN2)/β-actin from the Western blotting were quantified by grayscale scan; (**F**) immunofluorescence observation of the plasma membrane localization of CSN2 in cells subjected to a 24-h treatment with 0.6 mM methionine. CSN2 (green), Hoechst (blue). The scale bar is 100 μm. Data were the means ± SEM from three independent experiments. All data were analyzed using ordinary one-way ANOVA multiple comparisons and Tukey’s multiple comparison test. Values with different superscripted lowercase letters indicate a significant difference (*p* < 0.05). The Met in the graphs all represent methionine.

**Figure 3 metabolites-13-01149-f003:**
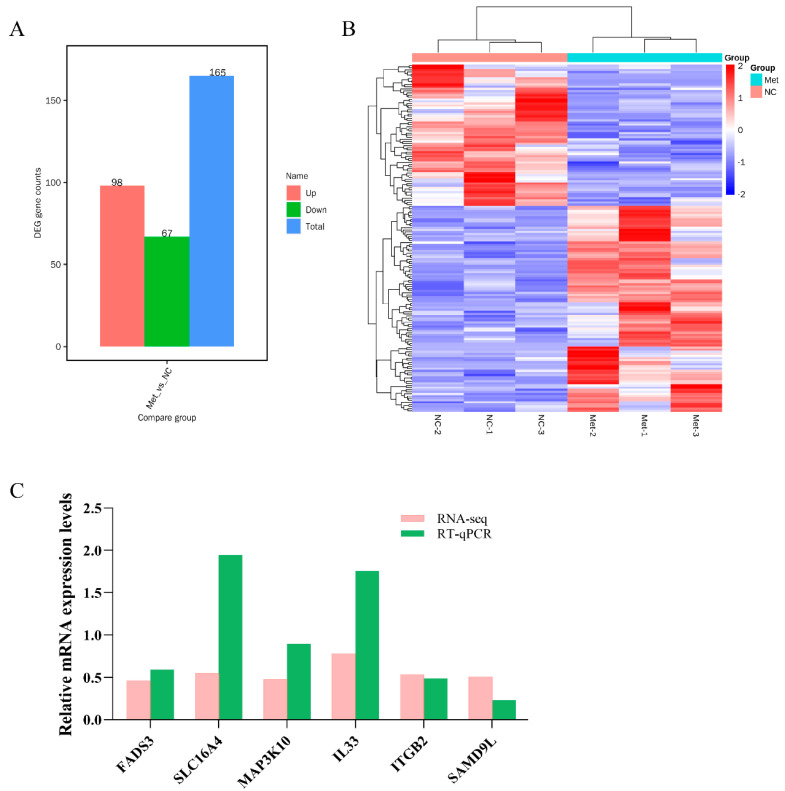
RNA-seq analysis in MCF-10A cells from control and methionine-supplemented groups: (**A**) Statistics of differentially expressed genes (DEGs); (**B**) DEG clustering (red indicates relatively high expression genes and blue indicates relatively low expression genes); (**C**) the fold change of DEG expression levels. The Met in the graphs all represent methionine. NC in the graph indicates the control group.

**Figure 4 metabolites-13-01149-f004:**
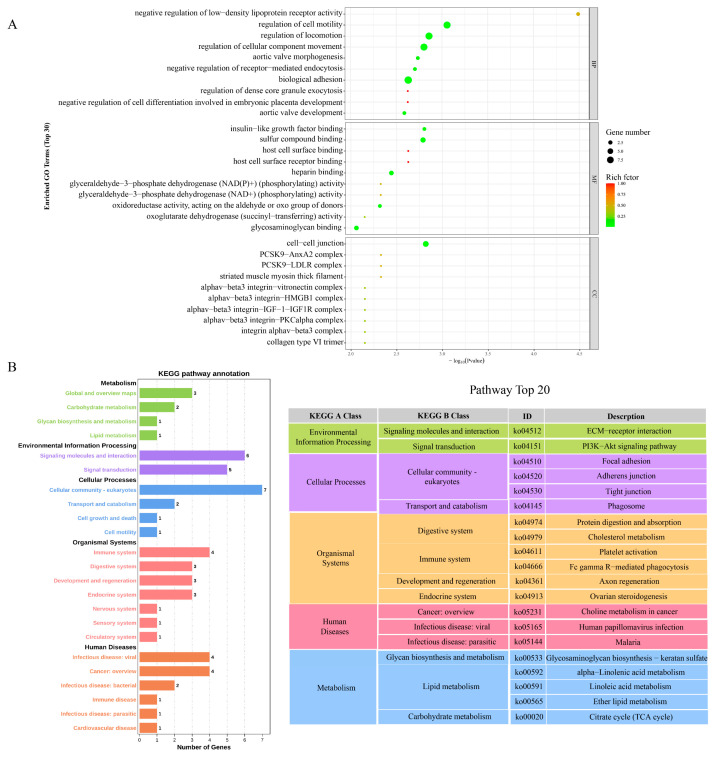
Differentially expressed gene (DEG) enrichment: (**A**) GO analysis of DEGs. From top to bottom: biological process (BP), molecular function (CF), cellular component (CC), vertical axis is descriptive information about a specific function; (**B**) KEGG enrichment map of DEGs.

**Figure 5 metabolites-13-01149-f005:**
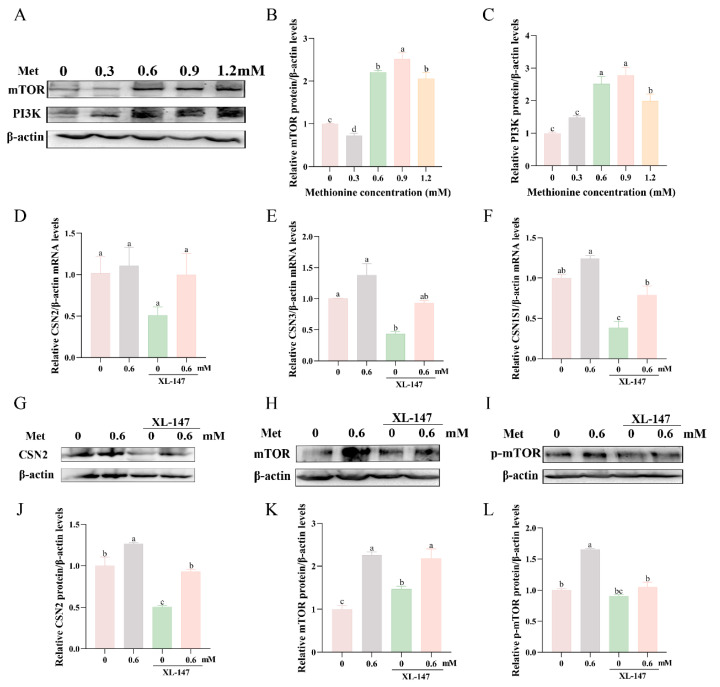
Effect of methionine on the expression of genes related to milk protein synthesis. MCF-10A cells were treated with different concentrations of methionine (0, 0.3, 0.6, 0.9, and 1.2 mM) for 24 h: (**A**) mTOR and PI3K protein levels were measured by Western blotting analysis; (**B**,**C**) relative folds of mTOR and PI3K protein levels (protein/β-actin) from the Western blotting were quantified by grayscale scan, effect of PI3K inhibition on the role of milk protein synthesis, MCF-10A cells were treated with XL-147 (477 nM) to inhibit PI3K; (**D**–**F**) effect of indicated treatment on the mRNA expression levels of CSN2, CSN3, and CSN1S1; (**G**–**I**) CSN2, mTOR, and p-mTOR protein levels were measured by Western blotting analysis; (**J**–**L**) relative folds of CSN2, mTOR, and p-mTOR protein levels (protein/β-actin) from the Western blotting were quantified by grayscale scan. Data were the means ± SEM from three independent experiments. All data were analyzed using ordinary one-way ANOVA multiple comparisons and Tukey’s multiple comparison test. Values with different superscripted lowercase letters indicate a significant difference (*p* < 0.05). The Met in the graphs all represents methionine.

**Table 1 metabolites-13-01149-t001:** Primers of real-time quantitative polymerase chain reaction (RT-qPCR).

Primer	Forward Sequences (5′-3′)	Reverse Sequences (5′-3′)
CSN2	GCAGGTCCCTCAGCCTATTC	ACAGCTCTCTGAGGGTAGGG
CSN1S1	AGGGCACCTAATCAGAGGGT	AATTGATGGCACTTACAGAACTGG
CSN3	AAATAGCCACCCACCCACTG	GCAGGAGCTGGTGTAGGTTC
FADS3	CCTGGCTCCTTATCTACCTCCT	GCTGGAAGTGGCGGAAGTT
SLC16A4	TCTCCTCAGTCAGTTAGCA	GAGCAAGCAGGTTAGTGAT
MAP3K10	AACCACAACCTCGCAGACA	TATTCATAGCCACGCCATACG
IL33	TACTCGCTGCCTGTCAACA	CAACACCGTCACCTGATTCAT
ITGB2	TATGTGGATGAGAGCCGAGAG	CCAGATGACCAGCAGGAGAA
SAMD9L	CAAGCAGGCAAGCACACTT	GTTAGACGACGCAGGAGGT
β-actin	AGACCTGTACGCCAACACAG	CGCTCAGGAGGAGCAATGAT

**Table 2 metabolites-13-01149-t002:** The top 15 DEGs in NC and Met.

Gene	Ensenble ID	NC	Met	Log2FC	Padj
ITGB3	ENSG00000259207	19.2	59.5	1.6322	0.00
IL33	ENSG00000137033	260.21	447.02	0.7808	0.00
SDCBP2	ENSG00000125775	226.52	355.9	0.6519	0.00
ITGA10	ENSG00000143127	144.82	217.96	0.5897	0.02
BTBD19	ENSG00000222009	189	282.59	0.5794	0.00
SPRED3	ENSG00000188766	173.43	254.18	0.5513	0.00
SLC16A4	ENSG00000168679	550.43	805.07	0.5485	0.00
NTSR1	ENSG00000101188	246.48	360.08	0.5459	0.00
DNAH1	ENSG00000114841	156.58	226.68	0.534	0.03
ITGB2	ENSG00000160255	1097.5	1589.83	0.5338	0.00
SAMD9L	ENSG00000177409	271	386.57	0.512	0.00
TPT1-AS1	ENSG00000170919	210.16	298.19	0.5043	0.00
TTLL3	ENSG00000214021	288.57	405.53	0.492	0.00
MAP3K10	ENSG00000130758	283.86	395.58	0.4786	0.00
FADS3	ENSG00000221968	827.16	1139.78	0.4619	0.00

DEGs, differentially expressed genes; FC, fold change; Padj, p.adjust; Met, methionine. NC, control group.

## Data Availability

The data presented in this study are available in the main article.

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
