# Peer review of "Methionine Promotes Milk Protein Synthesis via the PI3K-mTOR Signaling Pathway in Human Mammary Epithelial Cells"

_metabolites, 2023, doi:10.3390/metabo13111149_

Round 1

Reviewer 1 Report

Comments and Suggestions for Authors

This article addresses a little studied and very important topic that involves the synthesis of milk in the mammary gland in humans. The results add some incremental information on the mechanisms of milk production and the effects of methionine supplementation in the process.

The authors could deepen the discussion on the mechanisms by which proteins related to adhesion and motility, such as integrins, may be related to milk production. For that, I would like to recommend the citation of these two excellent reviews to support their data:

Slepicka PF, Somasundara AVH, Dos Santos CO. The molecular basis of mammary gland development and epithelial differentiation. Semin Cell Dev Biol. 2021 Jun;114:93-112. doi: 10.1016/j.semcdb.2020.09.014. Epub 2020 Oct 17. PMID: 33082117; PMCID: PMC8052380.

Twigger AJ, Engelbrecht LK, Bach K, Schultz-Pernice I, Pensa S, Stenning J, Petricca S, Scheel CH, Khaled WT. Transcriptional changes in the mammary gland during lactation revealed by single cell sequencing of cells from human milk. Nat Commun. 2022 Jan 28;13(1):562. doi: 10.1038/s41467-021-27895-0.

Minor points:

1. The authors could improve the quality of Figure 1A, the lines are very weak and difficult to see. If the results in Fig 1A were not significantly different (p> 0,05), so the text in lines 190-191 must be deleted. There were no changes in cell viability.

3. Fig. 2F – what was the time? 24h?

4. Fig. 1C , 2A,B,C,D, and the graphs on Fig 5: why is there a symbol of significance in the controls? Who are the controls being compared to? In all legends for figures, the authors should explain the meaning of lowercase letters – who is compared to whom?

5. Lines 194 and 195: Is there any explanation for the increase in cell proliferation and the sudden fall?

7. Fig. 2. What do the bell-shaped curves mean? The authors should discuss this effect.

8. Fig. 2F Please add zoom. Is CSN2 in the nucleus, cytoplasm or membrane, as stated in lines 219, 220 and 228? It is difficult to say for sure without co-localization experiments.

9. Line 263 and 264. There is something strange in this sentence. Please check.

10. Line 272. It should be “….the lactic synthesis in MCF-10A.

Comments on the Quality of English Language

The text is very well written, but I noticed a few mistyping errors

Reviewer 2 Report

Comments and Suggestions for Authors

The Manuscript by P. Li, X. Fang, G. Hao, X. Li, Y. Cai, Y. Yan, L. Zan, R. Yang, B. Liu “Methionine Promotes Milk Protein Synthesis via PI3K-mTOR Signaling Pathway in Human Mammary Epithelial Cells” describes the influence of methionine addition to nutrient medium on milk protein synthesis in MCF-10A cells and the study on biochemical basis of prolactogenic action of methionine. An important role of methionine in the production of milk and lactation is well-documented. Methionine and related compounds find a broad application as feed supplement in animal husbandry. So, the present study represents a detailed study of the biochemical mechanisms of action of methionine on human mammary epithelial cells. It is clearly demonstrated that methionine affects milk production not only as a structural component of casein and other proteins but also as an activator of PI3K-dependent signaling pathways. Attention to this aspect of prolactogenic action of methionine is undoubted advantage of the work. Several questions and remarks are given below.

1.    The rational of studying methionine and its effects on milk formation as the object of research has to be described more clearly in “Introduction” section. Why methionine was chosen but not other amino acids… The use of methionine, its derivatives and 2-hydroxy-4-(methylthio)butyric acid as feed supplements increasing milk production in animal husbandry should be mentioned and relevant references should be added.

2.    It remains unclear, why the most significant stimulation of casein production in the case of use methionine at concentration of 0.6 mmol/L is observed. The plausible reasons of the fact that an increase of methionine concentration to 0.9 mmol/L and higher results in the decrease of milk protein production should be mentioned in “Discussion” section.

3.    The references are listed not uniformly. Attention should be paid to refs. 23, 26, 31.

I believe that the manuscript by P. Li, X. Fang, G. Hao and co-authors deserves publishing in Metabolites after considering minor remarks listed above.
